# Bio-Distribution and Pharmacokinetics of Topically Administered γ-Cyclodextrin Based Eye Drops in Rabbits

**DOI:** 10.3390/ph14050480

**Published:** 2021-05-18

**Authors:** Martin Kallab, Kornelia Schuetzenberger, Nikolaus Hommer, Bhavapriya Jasmin Schäfer, Doreen Schmidl, Helga Bergmeister, Markus Zeitlinger, Aimin Tan, Phatsawee Jansook, Thorsteinn Loftsson, Einar Stefansson, Gerhard Garhöfer

**Affiliations:** 1Department of Clinical Pharmacology, Medical University of Vienna, 1090 Vienna, Austria; martin.kallab@meduniwien.ac.at (M.K.); nikolaus.hommer@meduniwien.ac.at (N.H.); doreen.schmidl@meduniwien.ac.at (D.S.); markus.zeitlinger@meduniwien.ac.at (M.Z.); 2Center for Medical Physics and Biomedical Engineering, Medical University of Vienna, 1090 Vienna, Austria; kornelia.schuetzenberger@meduniwien.ac.at (K.S.); priyajasmin@googlemail.com (B.J.S.); 3Christian Doppler Laboratory for Ocular and Dermal Effects of Thiomers, Medical University of Vienna, Währinger Gürtel 18-20, 1090 Vienna, Austria; 4Center for Biomedical Research, Medical University of Vienna, 1090 Vienna, Austria; helga.bergmeister@meduniwien.ac.at; 5Nucro-Technics, Toronto, ON M1H 2W4, Canada; tan@nucro-technics.com; 6Pharmaceutics and Industrial Pharmacy, Chulalongkorn University, Bangkok 10330, Thailand; phatsawee.j@chula.ac.th; 7Faculty of Pharmaceutical Science, University of Iceland, 107 Reykjavik, Iceland; thorstlo@hi.is; 8Department of Ophthalmology, University of Iceland, 101 Reykjavik, Iceland; einarste@landspitali.is

**Keywords:** γ-cyclodextrin, irbesartan, candesartan, multiple instillation, single instillation, nanoparticles

## Abstract

The purpose of this study was to evaluate the ocular pharmacokinetics, bio-distribution and local tolerability of γ-cyclodextrin (γCD) based irbesartan 1.5% eye drops and candesartan 0.15% eye drops after single and multiple topical administration in rabbit eyes. In this randomized, controlled study, a total number of 59 New Zealand White albino rabbits were consecutively assigned to two study groups. Group 1 (*n* = 31) received irbesartan 1.5% and group 2 (*n* = 28) candesartan 0.15% eye drops. In both groups, single dose and multiple administration pharmacokinetic studies were performed. Rabbits were euthanized at five predefined time points after single-dose administration, whereas multiple-dose animals were dosed for 5 days twice-daily and then euthanized 1 h after the last dose administration. Drug concentration was measured by using liquid chromatography-tandem mass spectrometry (LC-MS/MS) in the retinal tissue, vitreous humor, aqueous humor, corneal tissue and in venous blood samples. Pharmacokinetic parameters including maximal drug concentration (C_max_), time of maximal drug concentration (T_max_), half-life and AUC were calculated. To assess local tolerability, six additional rabbits received 1.5% irbesartan eye drops twice daily in one eye for 28 days. Tolerability was assessed using a modified Draize test and corneal sensibility by Cochet Bonnet esthesiometry. Both γCD based eye drops were rapidly absorbed and distributed in the anterior and posterior ocular tissues. Within 0.5 h after single administration, the C_max_ of irbesartan and candesartan in retinal tissue was 251 ± 142 ng/g and 63 ± 39 ng/g, respectively. In the vitreous humor, a C_max_ of 14 ± 16 ng/g for irbesartan was reached 0.5 h after instillation while C_max_ was below 2 ng/g for candesartan. For multiple dosing, the observed C_mean_ in retinal tissue was 338 ± 124 ng/g for irbesartan and 36 ± 10 ng/g for candesartan, whereas mean vitreous humor concentrations were 13 ± 5 ng/g and <2 ng/g, respectively. The highest plasma concentrations of both irbesartan (C_max_ 5.64 ± 4.08 ng/mL) and candesartan (C_max_ 4.32 ± 1.04 ng/mL) were reached 0.5 h (T_max_) after single administration. Local tolerability was favorable with no remarkable differences between the treated and the control eyes. These results indicate that irbesartan and candesartan in γCD based nanoparticle eye drops can be delivered to the retinal tissue of the rabbit’s eye in pharmacologically relevant concentrations. Moreover, safety and tolerability profiles appear to be favorable in the rabbit animal model.

## 1. Introduction

The renin-angiotensin system (RAS) plays a major role in the regulation of systemic hemodynamics [1]. Therefore, angiotensin receptor blockers (ARBs) and angiotensin converting enzyme (ACE) inhibitors are commonly used in the treatment of arterial hypertension, heart failure or kidney disease [2,3]. Beside its well described effects on systemic hemodynamic parameters, there is increasing evidence that the RAS also plays an important role in several physiological and pathological processes in the eye [4,5]. More specifically, it has been shown that ARBs as well as ACE inhibitors lower intraocular pressure (IOP) via intraocular angiotensin type I receptors [6,7]. In addition to this IOP lowering effect, ACE inhibitors and ARBs were shown to have beneficial effects on diabetic retinopathy progression and seem to be neuroprotective upon systemic administration, independent of their blood-pressure lowering properties [8,9,10,11,12].

Therefore, the development of eye drops containing ACE inhibitors or ARBs has been proposed in order to target the RAS directly in the eye to avoid systemic side effects such as hyperkalemia, hypotension or cough [13,14]. However, the low bioavailability of eye drop formulations containing ARBs and ACE inhibitors due to their lipophilicity and poor water solubility limit their therapeutic options after topical administration, in particular for the posterior segment [15].

To overcome these limitations, there is increasing interest in penetration enhancers, which allow drug formulations to enhance drug delivery across biological membranes that are otherwise impermeable or show limited permeability such as the cornea. Recently introduced cyclodextrin (CD) based drug formulations may enhance drug penetration and therefore increase bioavailability in the target tissues [16]. Indeed, previous evidence indicates that γCD based eye drops micro- and nanosuspensions can deliver drugs such as dexamethasone to both to the anterior and posterior pole both in rabbits and humans [17,18].

The current study aimed to evaluate the non-clinical pharmacokinetics and bio-distribution of γCD based competitive antagonists of the angiotensin II receptor irbesartan 1.5% eye drops and candesartan 0.15% eye drops after single and multiple instillations in rabbits. These formulation are aqueous microsuspensions in which the solid particles are metastable γCD/drug complexes that readily dissolve upon media dilution resulting in sustained high drug concentrations in the tear fluid [19].

Furthermore, local tolerability and potential toxic effects to the ocular surface were investigated in the same animal model. In summary, the data from the current study will serve as a basis for the further development of γCD based angiotensin II receptor antagonists in humans.

## 2. Results

### 2.1. Part 1: Biodistribution and Pharmacokinetics

Of the randomized rabbits, 58 out of 59 completed the study until the scheduled euthanasia. One rabbit had to be excluded before the drug administration due to genital lesions. The weights of rabbits at the time of arrival and before the randomization did not differ significantly between groups (not shown).

#### 2.1.1. Single-Dose Ocular Bio-Distribution

For both substances, the concentration gradient of calculated C_max_ values in the four analyzed tissues was as follows: corneal tissue > retinal tissue > aqueous humor > vitreous humor. Table 1 summarizes the C_max_ of the two different substances in the posterior eye segment (retinal tissue and vitreous humor) together with aqueous humor and corneal tissue results which have recently been published separately [20].

Drug exposure of the untreated control eye was generally low, with retinal tissue C_max_ of 23 ± 34 ng/g (irbesartan) and <2 ng/g (candesartan) and both vitreous humor maximum concentrations below the respective lower quantification limit. Control eye C_max_ of all tissues are presented in Table 1 together with the study eye C_max_.

#### 2.1.2. Single-Dose PK-Profiles for Retinal Tissue and Vitreous Humor

Retinal tissue PK-profiles of both compounds in both eyes and tabulations of the corresponding numerical values are shown in Figure 1 and Table 2, respectively. In the treated study eye T_max_ in retinal tissue was 0.5 h for both substances with a calculated T_1/2_ of 3.32 h (irbesartan) and 5.39 h (candesartan). Retinal tissue AUC of irbesartan was 941 ng/g*h while the corresponding AUC of candesartan reached 210 ng/g*h. In relative numbers irbesartan retinal tissue AUC was around 4.5 times higher than the corresponding candesartan AUC. Vitreous humor drug exposure was low for both substances. Study eye concentrations were above the respective lower limit of quantification at one time point for irbesartan (14 ± 16 ng/g at T0.5) and at no timepoint for candesartan (Table 2).

#### 2.1.3. Multiple Dose Bio Distribution

The mean concentrations (C_mean_) in aqueous humor, corneal tissue, retinal tissue and vitreous humor of both eyes 1 h ± 30 min after the last dose of irbesartan or candesartan are presented in Table 3. Similarly to the single dose results, irbesartan eye drops reached nominally higher overall concentrations than candesartan eye drops in the treated eyes. The study eye C_mean_ of retinal tissue was 338 ± 124 ng/g for irbesartan and 36 ± 10 ng/g for candesartan while the vitreous humor mean concentrations were 13 ± 5 ng/g (irbesartan) and <2 ng/g (candesartan).

#### 2.1.4. Blood-Plasma PK

The highest plasma concentrations of both irbesartan (C_max_ 5.64 ± 4.08 ng/mL) and candesartan (C_max_ 4.32 ± 1.04 ng/mL) were reached 0.5 h (T_max_) after single administration. The plasma half-life and AUC for irbesartan were 17.60 h and 17.64 (ng/mL)*h, respectively. For candesartan a T_1/2_ of 9.74 h and AUC of 8.75 (ng/mL)*h were calculated. After the last dose of one of the study drugs mean plasma levels (C_mean_) of 4.21 ± 1.59 ng/mL for irbesartan and 4.78 ± 1.48 ng/mL for candesartan were observed.

### 2.2. Part 2: Toxicity and Local Tolerability

All six rabbits completed the study according to the protocol. No reduction in corneal sensitivity was observed during the course of the study. Further, no signs of discomfort became evident during the whole study period. In detail, no conjunctival edema, conjunctival redness, abnormal secretion, corneal opacity or iris involvement could be detected using the modified Draize test scoring system. Dilated fundoscopy was unremarkable at all examinations on study days 1, 15 and 29 in all animals. All assessed vital signs (body weight, heart rate, respiratory rate, body temperature) remained within the normal range throughout the study.

## 3. Discussion

Irbesartan and candesartan are specific competitive antagonists of the angiotensin II receptor (AT1 subtype) and have been proposed as a potential treatment option for ocular diseases. Here, we report high ocular tissue concentrations combined with low systemic drug exposure after topical administration for both γCD based candesartan and irbesartan eye drops. Further, both drug formulations were well tolerated on the ocular surface and did not cause severe adverse events in the selected rabbit animal model.

CD based pharmaceutical technologies have significantly evolved since the early 80’s. Chemically speaking, CDs are water-soluble cyclic oligosaccharides which can form nanoparticles [16]. As such, a large variety of CDs and their derivatives have been developed, for example, 2-hydroxypropyl-βCD, sulfobutyl ether βCD and 2-hydroxypropyl-γCD, each with different chemical and physical properties [21]. As their major principle of action, CDs are capable of forming guest–host inclusion complexes, whereby lipophilic drugs with poor aqueous solubility can be entrapped in the hydrophobic cavity without being covalently bound. Thus, aqueous solubility is improved due to the hydrophilic properties of the external surface of the CD molecule [18,21]. CDs enhance drug permeation from the tear fluid by enhancing drug permeation through the mucus layer and, thus, increasing the availability of dissolved drug molecules immediate to the lipophilic membrane barriers (i.e., cornea and conjunctiva).

The use of CDs in topical ocular drug formulations has been recently reported to improve the solubilization of various substances including corticosteroids, medications for glaucoma treatment and immunosuppressive agents [17,22,23]. In particular, CDs enhance ocular permeability of poorly soluble drugs through the lipophilic corneal epithelial membrane and increase their bioavailability in the anterior chamber [24]. The data of our study indicates that this holds also true for the two formulations tested in our study.

As such, our data show an anterior chamber concentration with a C_max_ of 121 ng/g for irbesartan and 30 ng/g for candesartan, respectively. Further, our results indicate that at the level of the retina drug concentration reached 251 ± 143 ng/g for irbesartan and 63 ± 39 ng/g for candesartan with a maximum effect 30 min after single administration. In this context, IC50 for candesartan and irbesartan is reported to be between 0.4 nM and 1.3 nM, respectively with candesartan showing a minutely but consistently higher affinity [25,26]. Thus, assuming a molar weight of 428.54 g/mol for irbesartan and 440.45 g/mol for candesartan, retinal tissue C_mean_ in our multiple dose experiments were 789 nM (irbesartan) and 82 nM (candesartan) while single dose C_max_ values correspond to 586 nM (irbesartan) and 144 nM (candesartan), exceeding the IC50 concentration about 500-fold for irbesartan and 100-fold for candesartan. This suggests that with both candesartan and irbesartan eye drops, pharmacologically relevant concentrations can be reached in the back of the eye. Our data also indicates a higher biodistribution of irbesartan in the ocular tissues, which may be related to the higher drug load of the irbesartan γCD -based nanoparticles. Additional studies are needed to further investigate this issue. Further, significant concentrations of the study drugs in all assessed tissues and compartments were found in the study eye, while systemic absorption was relatively low. This is both reflected in the low plasma concentrations and in the finding that almost no evidence of the drug was found in the control eye.

Our finding that γCD based formulations may be a potential option for increased drug delivery to various tissues of the eye including the posterior pole is also supported by previously published studies using a variety of different CD based drug formulations. Indeed, several lipophilic drugs have been tested using CD formulations to achieve higher drug concentrations in the posterior and anterior pole of the eye [27,28,29,30,31]. As such, Larsen and colleagues showed that different CD based formulations of prednisolone are able to improve the drug solubility, increase drug concentration and in turn, ameliorate bioavailability [32]. Further, an in vivo study in rabbits by El-Gawad and colleagues reported that eye drop formulations containing CD/econazole complexes showed higher ocular bioavailability than econazole alone, indicated by higher AUC, C_max_, and relative bioavailability values [33]. Comparable observations have been reported by Abdelkader et al., where a diclofenac complex with CD has been reported to feature a higher transfer rate through the cornea compared to free drug administration [34].

Our data show that γCD irbesartan and candesartan nanoparticle eye drops are well tolerated on the ocular surface. In particular, our 4 weeks ocular tolerance study showed an excellent safety profile with no signs of adverse effects on the ocular surface or deeper tissues of the eye. Again, these results are in keeping with previous studies on γCD based eye drops. On the report of several studies, the natural γCD has a favorable toxicological profile, high solubility and good complexation capacity when compared with other CD derivatives [35]. In particular, previous data from animal models indicate that formulations of γCD/cyclosporine are tolerated well on the eye up to 3 months of administration [23].

Although a variety of different competitive antagonists of the angiotensin II receptor are in principle available for clinical use, the decision to use irbesartan and candesartan based γCD eye drops in our study was based on pre-formulation experiments. Data from these pre-formulation studies showed that irbesartan had the highest affinity for the γCD cavity among the tested compounds which included telmisartan, olmesartan, irbesartan and candesartan. Further, although the complexing efficiency of the candesartan γCD complex was lower, candesartan has been chosen as potential second drug for further development because of its good receptor affinity and potent in-vivo drug effects.

The present study also has some limitations that warrant further discussion. First, given that drug concentration measurement required post-mortem enucleation and dissection of the eye, only five animals are available per timepoint. A larger number of animals would certainly allow for a higher precision and reliability of calculated PK parameters but must be outweighed against ethical and animal welfare considerations. However, we feel that the current data is solid enough to allow valid conclusions regarding the tissue PK and tolerability of the surface. In the future, more sophisticated methods such as ocular microdialysis as described previously may provide approaches to help reduce animal numbers [36]. Other methods for molecular imaging to determine biodistribution, such as position emission tomography (PET), single-photon emission computed tomography (SPECT), magnetic resonance imaging (MRI) or Confocal Raman spectroscopy are also currently under development and may provide non-invasive alternatives to assess biodistribution in vivo [37,38]. Secondly, our tolerability data is limited to a twice-daily administration at a maximum time span of 28 days. Therefore, long-term adverse effects becoming evident after that time point cannot be excluded.

In conclusion, the data provided in this study demonstrate that γCD based irbesartan 1.5% and candesartan 0.15% eye drops deliver a significant amount of drug substance to several ocular tissues including the posterior pole of the eye. It confirms that it is possible to achieve effective trans-ocular delivery of lipophilic drugs, such as irbesartan and candesartan, by producing a complex with CD nanoparticles. Together with the good tolerability profile, the results suggest that γCD may be useful for the formulation of a variety of poorly water-soluble drugs that have not been available as commercial topical eye drops.

## 4. Materials and Methods

### 4.1. Test Animals

Female New Zealand White (NZW) albino rabbits were used in this study. All test animals were acclimatized at least one week prior to the study and housed under the following standardized conditions: artificial day tonight rhythm 12:12, room temperature 20 °C ± 2 °C and humidity 55 ± 5% with food and water ad libitum. This study was carried out in compliance with the European Directive 2010/63/EU, the Austrian Tierversuchsgesetz (BGBL I Nr. 114/2012) and the Austrian Tierversuchsverordnung (BGBL II Nr. 522/2012) (approval numbers: BMBWF-66.009/0206-V/3b/2018, BMBWF-66.009/0135-V/3b/2019, approved on 13 July 2018, 08 April 2019) and in accordance with the Good Scientific Practice Guidelines of the Medical University Vienna.

### 4.2. Study Drug Compounds

In the CD nanoparticle suspensions, 1.5% irbesartan and 0.15% candesartan as described previously in detail [39], were applied as eye drops in this study. Pre-formulation studies with different candesartan (0.1–0.2%) and irbesartan (1.0–2.0%) concentrations were performed to find the optimal drug concentration for the animal experiments. Based on this data, it was concluded that the optimal formulations were 0.15% candesartan and 1.5% irbesartan eye drops, both of which exhibited good physicochemical parameters as well as physical and chemical stability. Details regarding stability are given in Appendix A. Eye drops were stored protected from light at room temperature (25–30 °C).

### 4.3. Experimental Paradigm

#### 4.3.1. Part 1: Bio Distribution and Pharmacokinetics

A total number of 59 rabbits were included in this part of the study. All rabbits were consecutively assigned to one of the two study groups. In group 1, 26 rabbits received a single dose of 1.5% Irbesartan eye drops whereas 5 rabbits received multiple doses of the same irbesartan eye drops. Group 2 animals were dosed with 0.15% candesartan eye drops, while 23 rabbits received a single dose and 5 rabbits received multiple doses of the eye drops. Single dose rabbits of both groups were additionally randomized to one of five euthanasia time points: 0.5 h ± 5 min, 1.5 h ± 5 min, 3 h ± 10 min, 6 h ± 15 min or 12 h ± 30 min after instillation. In all animals, 35 μL of the study drug were instilled in the conjunctival sac of the right eye with a calibrated pipette using single-use dispensers. Single dose animals were instilled only once while multiple dose animals were instilled twice daily for a total of 5 days. In all animals only the right eye was treated. To assess drug concentrations, eyes were enucleated at the end of the study period. Before enucleation, rabbits were euthanized in deep anesthesia (ketamine 60 mg/kg, xylazine 16 mg/kg, s.c.) by an intravenous over-dose of pentobarbitone sodium (300 mg/kg i.v.) and enucleation was performed immediately after euthanasia. Single dose animals were enucleated at one of the five specified time-points. Multiple dose animals were euthanized 1 h ± 30 min after the last drug administration. From each animal both eyes were enucleated and stored at −80 °C. For separation of the different ocular tissues, the eyeball was removed from −80 °C storage and corneal tissue, aqueous humor, vitreous humor and retinal/choroidal tissue were dissected. The tissues were collected separately in individual tubes, which were weighed and then stored at −80 °C.

##### Blood Sampling

A venous blood sample of 2 mL was collected from the saphenous vein before the (first) drug administration and prior to euthanasia to determine systemic drug exposure at baseline and after drug administration. The blood was centrifuged for 10 min at a temperature of 10 °C and a speed of 3000 rpm. The resulting supernatant was separated and stored in a tube at −80 °C until further processing.

##### Determination of Drug Concentrations

Both, the eye-tissue and plasma samples were shipped to Nucro-Technics (Nucro-Technics, 2000 Ellesmere Road, Unit #16, Scarborough, ON, Canada) for quantitative analysis. For this purpose, a new liquid chromatography/tandem mass spectrometry method for simultaneous quantification of candesartan and irbesartan in rabbit eye tissues including cornea, aqueous humor, vitreous body and retina was developed. The exact method and validation are described in detail elsewhere [40].

#### 4.3.2. Part 2: Local Tolerability

Local tolerability and potential adverse effects of 1.5% irbesartan were assessed in a group of 6 rabbits. One drop equaling 35 µL of 1.5% irbesartan was instilled twice daily into the conjunctival sac of the right eye of each rabbit at 8:00 a.m. ± 1 h and 4:00 p.m. ± 1 h for a total duration of 28 days using a standard eye drop dispense bottle provided by the manufacturer. The left eye was left untreated and served as control. A five-item modified Draize test [41] scoring system was used to assess local tolerability of the study drug. It included conjunctival edema (chemosis), redness in conjunctiva, secretion, corneal opacity and iris involvement. Ocular assessment was done before and after the first dose on the first study day, on the second study day and thereafter on a weekly basis using a hand-held slit lamp. Further, dilated fundoscopy (indirect ophthalmoscopy) was performed before the first dose and afterwards every second week (study day 15 and 29) using topical 0.5% tropicamide eye drops for pupil dilation. Corneal sensibility was quantified using a Cochet–Bonnet esthesiometer (Luneau Ophthalmologie, Chartres Cedex, France) [42] at the same time points as fundoscopy. Heart rate, respiratory rate, body temperature and body weight were monitored in regular intervals throughout the study. After 28 days, animals were euthanized in anesthesia as described above.

### 4.4. Statistical Analysis

This study was designed to assess pharmacokinetics and local tolerability, no formal sample size calculation has been performed. Descriptive statistics was used for all quantitative data to describe drug concentrations after the single and multiple dosing in the different tissues. Using the drug concentrations measured in the tissue samples maximal drug concentration (C_max_), time of maximal drug concentration (T_max_), half-life time (T_1/2_) and area under the curve (0 h–12 h; AUC) were calculated. For pharmacokinetic calculations Phoenix WinNonlin 8.1 (Certara, Princeton, NJ, USA) was used, and the statistical analysis was done using CSS Statistica 6.0 (Tulsa, OK, USA).

To facilitate calculation of the PK parameters, a plausibility check was performed and values below the lower limit of quantification (BLOQ; plasma: <0.05 ng/mL; tissue: <5 ng/g (irbesartan), <2 ng/g (candesartan)) were set to 0. Moreover, one 6 h irbesartan plasma concentration, which differed more than 10-fold from the mean of the other irbesartan 6 h concentrations (65.24 ng/mL vs. 1.16 ng/mL) was defined as an outlier and was therefore not taken into consideration for the presented PK profile. All baseline plasma sample values were set to zero.

## Figures and Tables

**Figure 1 pharmaceuticals-14-00480-f001:**
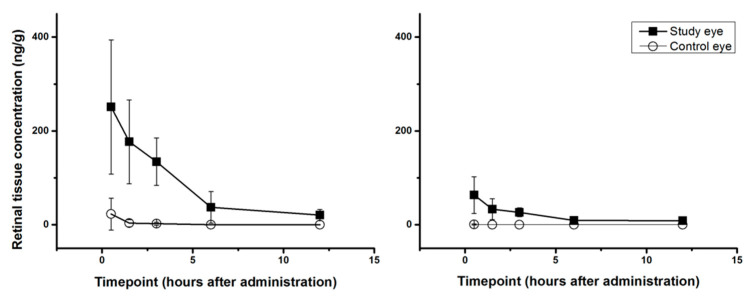
Retinal tissue concentrations of irbesartan (**left**) and candesartan (**right**) in the study and control eye. Data are shown as means ± SD.

**Table 1 pharmaceuticals-14-00480-t001:** C_max_ after a single dose of 1.5% irbesartan or 0.15% candesartan calculated for the four analyzed tissues in the study (SE) and control eye (CE), RT: retinal(choroidal) tissue, VH: vitreous humor, AH: aqueous humor, CT: corneal tissue, N/A: concentrations in all samples at all timepoints were below the lower limit of quantification and therefore T_max_ was not determinable. C_max_ are shown as means ± SD.

Study Drug-Tissue	C_max_ SE (ng/g)	T_max_ SE (h)	C_max_ CE (ng/g)	T_max_ CE (h)
Irbesartan-RT	251 ± 143	0.5	23 ± 34	0.5
Irbesartan-VH	14 ± 16	0.5	<5	0.5
Irbesartan-AH	121 ± 69	3	<5	0.5
Irbesartan-CT	3663 ± 988	1.5	49 ± 85	1.5
Candesartan-RT	63 ± 39	0.5	<2	0.5
Candesartan-VH	<2	0.5	<2	N/A
Candesartan-AH	30 ± 14	3	<2	N/A
Candesartan-CT	3504 ± 801	1.5	2 ± 4	0.5

C_max_ for all analyzed ocular tissues of the study eye was nominally higher in the 1.5% irbesartan group as compared to the 0.15% candesartan group. For retinal tissue the C_max_ were 251 ± 143 ng/g (irbesartan) and 63 ± 39 ng/g (candesartan) while vitreous humor C_max_ were 14 ± 16 ng/g (irbesartan) and <2 ng/g (candesartan), respectively.

**Table 2 pharmaceuticals-14-00480-t002:** Retinal tissue (RT) and vitreous humor (VH) concentrations of irbesartan and candesartan at each time point in the study and the control eye. Data are shown as means ± SD in ng/g.

Study Drug	T0.5	T1.5	T3.0	T6.0	T12.0
**1.5% irbesartan**					
RT-Study eye (ng/g)	251 ± 143	177 ± 89	134 ± 51	37 ± 34	21 ± 2
RT-Control eye (ng/g)	23 ± 34	<5	<5	<5	<5
VH-Study eye (ng/g)	14 ± 16	<5	<5	<5	<5
VH-Control eye (ng/g)	<5	<5	<5	<5	<5
**0.15% candesartan**					
RT-Study eye (ng/g)	63 ± 39	33 ± 22	27 ± 9	9 ± 4	9 ± 3
RT-Control eye (ng/g)	<2	<2	<2	<2	<2
VH-Study eye (ng/g)	<2	<2	<2	<2	<2
VH-Control eye (ng/g)	<2	<2	<2	<2	<2

**Table 3 pharmaceuticals-14-00480-t003:** Tissue concentrations of irbesartan and candesartan after multiple doses of 1.5% irbesartan or 0.15% candesartan in the study (SE) and the control eye (CE). Data are shown as means (C_mean_) ± SD.

	1.5% Irbesartan		0.15% Candesartan	
Tissue Samples	SE (ng/g)	CE (ng/g)	SE (ng/g)	CE (ng/g)
RT	338 ± 124	7 ± 8	36 ± 10	<2
VH	13 ± 5	<5	<2	<2
AH	231 ± 68	<5	70 ± 22	<2
CT	9027 ± 2156	39 ± 30	7468 ± 908	<2

## Data Availability

The data presented in this study are available on reasonable request from the corresponding author.

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
