# Peer review of "Bio-Distribution and Pharmacokinetics of Topically Administered γ-Cyclodextrin Based Eye Drops in Rabbits"

_pharmaceuticals, 2021, doi:10.3390/ph14050480_

Round 1

Reviewer 1 Report

The research article entitled Bio-distribution and Pharmacokinetics of Topically Administered g-Cyclodextrin Based Eye Drops in Rabbits shows the biodistribution and ocular toxicity of g-cyclodextrin (gCD) based irbesartan 1.5% eye drops and candesartan 0.15% eye drops after single and multiple topical administration in rabbit eyes. This is a valuable research. Here are some concerns and related suggestions to improve the article;

  • There are two drugs formulated with cyclodextrin as eye drops: irbesartan and candesartan. Authors’ previous study (doi:10.3109/10837450.2014.910811.) is referred for the particle preparation but only covers the pharmacokinetic data for irbesartan. It’s required to give detailed experimental findings for loading / stability / shelf-life / degradation of candesartan loaded cyclodextrins. Also, an explanation on why these drug molecules have been tested as opposed to each other is required.
  • What is the reason for using 0.15% candesartan and 1.5% irbesartan? Is there any other concentration tested? Any relevant background studies should be shared.
  • Irbesartan-cyclodextrin formulation shows higher biodistribution on all ocular tissues tested. What would be the reason for a a better biodistribution compared to candesartan? I would consider the drug loading into cyclodextrin, retention time differences and fast drug release kinetics. These should be discussed.
  • The study was conducted with 59 animals. What was the sample size determined to obtain significant data? This data should be in the study design.

Author Response

We thank the reviewer for their thoughtful comments on our manuscript. In the following, we respond to the comments of the reviewer on a point to point basis. For the convenience of the reviewers, changes in the revised manuscript are marked in yellow color. We hope that the revised version of our manuscript will be suitable for publication.

Reviewer comments:

The research article entitled Bio-distribution and Pharmacokinetics of Topically Administered g-Cyclodextrin Based Eye Drops in Rabbits shows the biodistribution and ocular toxicity of g-cyclodextrin (gCD) based irbesartan 1.5% eye drops and candesartan 0.15% eye drops after single and multiple topical administration in rabbit eyes. This is a valuable research. Here are some concerns and related suggestions to improve the article;

Comment: There are two drugs formulated with cyclodextrin as eye drops: irbesartan and candesartan. Authors’ previous study (doi:10.3109/10837450.2014.910811.) is referred for the particle preparation but only covers the pharmacokinetic data for irbesartan. It’s required to give detailed experimental findings for loading / stability / shelf-life / degradation of candesartan loaded cyclodextrins. Also, an explanation on why these drug molecules have been tested as opposed to each other is required.

Answer: We thank the reviewer for this comment. The decision to use irbesartan and candesartan based cyclodextrin eye drops in the animal studies were based on pre-formulation experiments. Data from these pre-formulation studies showed that irbesartan had the highest affinity for the gCD cavity among the tested compounds which included telmisartan, olmesartan, irbesartan and candesartan. Further, although the complexing efficiency of the candesartan/gCD complex was lower, candesartan has been chosen as second drug for further development because of its good receptor affinity and potent in-vivo drug effects. As suggested by the reviewer, we have mentioned this in the revised version of the manuscript.

The data of the current study will serve as a basis to decide if one or both of the gCD formulations will be further developed for human application. We have added this to the revised version of the manuscript. Additionally, as requested by the reviewer we have added data regarding the stability data as a supplemental table (S-1) to the revised version of the manuscript.

Comment: What is the reason for using 0.15% candesartan and 1.5% irbesartan? Is there any other concentration tested? Any relevant background studies should be shared.

Answer: Following the suggestion of the reviewer, we have explained the dose rationale in more detail in the method section of the revised manuscript. During the pre-formulation studies, different candesartan (concentrations: 0.1-0.2%) and irbesartan (concentrations: 1.0-2.0%) eye drop formulations were developed and investigated. Based on the data of these studies, it was concluded that the optimal formulations were 0.15% candesartan and 1.5% irbesartan eye drops, both of which exhibited better physicochemical parameters as well as physical and chemical stability compared to the other drugs tested. We have added this to the revised version of our manuscript.

Comment: Irbesartan-cyclodextrin formulation shows higher biodistribution on all ocular tissues tested. What would be the reason for a better biodistribution compared to candesartan? I would consider the drug loading into cyclodextrin, retention time differences and fast drug release kinetics. These should be discussed.

Answer: We thank the reviewer for this comment. As correctly stated by the reviewer, the reason for the better biodistribution of irbesartan is most probably related to higher drug load of irbesartan gCD-based nanoparticles. We have mentioned this in the revised version of our manuscript.

Comment: The study was conducted with 59 animals. What was the sample size determined to obtain significant data? This data should be in the study design.

Answer: As this is a PK and toxicology study and data are presented in a descriptive manner, no formal sample size calculation has been performed prior to the study. We have mentioned this in the revised version of the manuscript.

Reviewer 2 Report

The manuscript ¨Bio-distribution and Pharmacokinetics of Topically Administered γ-Cyclodextrin Based Eye Drops in Rabbits" shows the first steps for a further and extensive research in the area.

I would like to add some minor corrections and suggestions:

Sentence 17. Abstract: "The" should not be bold. 

In order to better illustrate the local toxicity and tolerability, the authors should include some images of the rabbits with at least 3 different stages of the study.

Apart from that, an as it is mentioned in the manuscript, the n for a better an concise conclusion should be higher, and maybe it would be interesting to try to find a non invasive way of measuring PK/PD.

Author Response

We thank the reviewer for their thoughtful comments on our manuscript. Please find our response below:

Comment: Sentence 17. Abstract: "The" should not be bold.

Answer: We corrected this formatting issue.

Comment: In order to better illustrate the local toxicity and tolerability, the authors should include some images of the rabbits with at least 3 different stages of the study.

Answer: Unfortunately, we did not obtain images from the rabbits, since this was not part of the study protocol.

Comment: Apart from that, an as it is mentioned in the manuscript, the n for a better an concise conclusion should be higher, and maybe it would be interesting to try to find a non invasive way of measuring PK/PD.

Answer: We totally agree with the reviewer. As stated on page 7, line 221, new methods like ocular microdialysis might be helpful in sparing animals. We also added information about new, non-invasive methods that are currently under study.